# *Exaiptasia pallida* Infection Model Reveals the Critical Role of *Vibrio parahaemolyticus* T3SS Virulence Factors in Its Pathogenicity for Sea Anemones

**DOI:** 10.3390/toxins17040175

**Published:** 2025-04-02

**Authors:** Alexandre Perrone, Estelle Bonnet, Anna Soone, Laurent Boyer, Francois Seneca

**Affiliations:** 1Centre Scientifique de Monaco, Département de Biologie Médicale, 8 Quai Antoine 1er, 98000 Monaco, Monaco; anna.soone@gmail.com; 2Centre Méditerranéen de Médecine Moléculaire, Inserm U1065, Université Côte d’Azur, 06204 Nice, Francelaurent.boyer@univ-cotedazur.fr (L.B.); 3LIA ROPSE, Laboratoire International Associé, Centre Scientifique de Monaco, Université Côte d’Azur, 06103 Nice, France

**Keywords:** host–pathogen, virulence factors, secretion system, Vibrio, sea anemone, pathogenicity

## Abstract

*Vibrio parahaemolyticus* is the leading cause of seafood-borne gastroenteritis. While its interaction with edible marine animals is well known, its impact on non-edible hosts remains under-explored. Using the sea anemone *Exaiptasia pallida*, we investigated *Vibrio parahaemolyticus* pathogenicity and the role of the Type III Secretion System (T3SS). In vivo infections with a 10^7^ CFU/mL inoculum of *V. parahaemolyticus* induced a 50% mortality rate after 7 days (LC50). Using isogenic mutant strains of *V. parahaemolyticus* with impaired key regulatory components of T3SS, impT3SS1 (CAB2), and impT3SS2 (CAB3), we demonstrated that disruption of T3SS1 significantly reduced anemone mortality. Next, we observed a time-dependent downregulation of T3SS1 effectors (VPA0450, VopQ, VopS) after 3 h and 6 h in the presence of the sea anemone, contrasting with the T3SS2-dependent VopC increased expression after 6 h. Further results support the capacity of *V. parahaemolyticus* to sense host-derived chemical cues and adjust its virulence strategies accordingly. Collectively, our findings broaden the understanding of *V. parahaemolyticus* O3:K6 as a pathogen for cnidarians and provide evidence of a major role for the T3SS1 effectors in this emerging model of host–pathogen interactions.

## 1. Introduction

*Vibrio parahaemolyticus* is a public health concern causing approximately 50,000 infections per year in the USA [1]. Since the late 1990s, its prevalence has expanded making it the leading cause of seafood-borne gastroenteritis worldwide [2,3,4]. This Gram-negative, halophilic bacterium frequently inhabits estuarine and coastal waters, colonizing sediments, plankton, and various aquatic animals, notably shellfish such as oysters, clams, and crabs as well as corals [5,6,7,8,9]. The US Centers for Disease Control and Prevention (CDC) has emphasized the substantial public health and food safety implications of *V. parahaemolyticus*, underscoring the need for effective monitoring and control measures [10]. Human infections often stem from consuming raw or undercooked seafood. Consequently, *V. parahaemolyticus* has a significant economic impact by driving up healthcare expenses and by reducing seafood productivity, as well as reducing consumer seafood preferences [11,12,13,14]. While *V. parahaemolyticus* infection in various aquaculture species has been extensively investigated, its impact on non-edible marine animals, such as cnidarians, has been largely ignored.

Beyond its impact on human health, *V. parahaemolyticus* occurs in both healthy and diseased corals, underscoring its diverse ecological roles. For instance, it can be a dominant member of the bacterial community in coral mucus and tissues [8,9], which suggests possible mutualistic interactions by providing nutrients or secondary metabolites. However, *V. parahaemolyticus* has also been implicated in coral diseases, particularly when environmental stressors such as elevated temperatures can promote the expression of virulence factors and adhesion to plastics [9,15,16]. In fact, strains closely related to *V. parahaemolyticus* have been identified as causative agents of yellow blotch/band disease in Caribbean corals [15,17], emphasizing that while this bacterium can be benign or even beneficial in stable conditions, it may shift to a pathogenic state under stress [18,19]. Moreover, the positive feedback loop between the increasing cases of human vibriosis and the contamination of marine organisms via human effluent warrants the study of zoonotic-driven vibriosis in a cnidarian species.

We previously explored the host transcriptomic activity of a model cnidarian species, the sea anemone *Exaiptasia pallida,* during infection with the pathogenic *V. parahaemolyticus* strain O3:K6 to identify genes involved in innate immunity [20]. This work revealed that genes homologous to those in human innate immunity are important to the anemone’s defense, suggesting that fundamental immune signaling pathways have been conserved for over 600 million years. Nonetheless, additional research is required to determine the impact of *V. parahaemolyticus* effectors during *E. pallida* infection and to identify whether they are responsible for driving pathogenic interactions with cnidarians.

*V. parahaemolyticus* has evolved sophisticated molecular mechanisms to infect marine organisms, including the production of several virulence factors injected in the host via dedicated secretion systems. A major virulence factor is the thermostable direct hemolysin (TDH), which disrupts host cell membranes and leads to cell lysis and tissue damage [1]. Meanwhile, the T3SS, a type of molecular syringe, allows *V. parahaemolyticus* to inject effector proteins directly into host cells, manipulating cellular functions to favor bacterial survival. Two distinct T3SS gene clusters (T3SS1 and T3SS2) are differentially associated with pathogenic outcomes, with T3SS2 particularly involved in enterotoxicity and tissue invasion [21].

In this study, we investigated how the pathogenic *V. parahaemolyticus* strain O3:K6 interacts with the sea anemone *E. pallida*. We demonstrated that *V. parahaemolyticus* can actively kill *E. pallida*, determining an inoculum that led to 50% mortality over one week. We then assessed the role of the *V. parahaemolyticus* type III secretion system (T3SS) using isogenic strains mutated for a T3SS syringe expression regulator gene. Finally, we evaluated the expression of specific effector genes in the presence or absence of live anemones and investigated whether chemical cues or molecules in filtered anemone lysate induce bacterial pathogenicity. Collectively, our findings shed new light on the pathogenicity of *V. parahaemolyticus* in a cnidarian host and broaden the fundamental understanding of the role of virulence factors in the sea anemone *E. pallida* model of infection.

## 2. Results

### 2.1. E. pallida Sensitivity to V. parahaemolyticus

To determine the lethal concentration (LC50) of *V. parahaemolyticus* for *E. pallida*, we exposed 12 anemones to three bacterial concentrations: 10^6^, 10^7^, and 10^8^ CFU/mL.

After two days, anemones exposed to wild-type (WT) *V. parahaemolyticus* showed differences in their morphological responses in a dose-dependent manner (Figure 1A). At the lowest concentration: 10^6^ CFU/mL, anemones remained visually indistinguishable from control animals not exposed to bacteria. However, when exposed to 10^7^ CFU/mL, anemones displayed marked contraction of their tentacles and columns, although they were still responsive to bright light. In contrast, at the highest concentration: 10^8^ CFU/mL, all anemones detached from the bottom of the dish, showed a blob-like appearance, and lost responsiveness to both light and touch stimuli. Anemones reaching this stage were considered dead.

After one week post-exposure, anemones reveal a dose-dependent mortality (Figure 1B). While at 10^8^ CFU/mL all anemones died after 2 days, at 10^7^ CFU/mL, the mortality gradually reached 100% by day 8. Only 25% of mortality occurred within the same timeframe at 10^6^ CFU/mL; 50% of mortality was reached 7 days after exposure to 10^7^ CFU/mL bacteria. From these observations, the LC50 over a week was 10^7^ CFU/mL, making this dose optimal for further studies of *V. parahaemolyticus* pathogenicity in anemones.

### 2.2. T3SS1 Involvement in E. pallida’s Mortality

To assess the role of Type III Secretion Systems (T3SS) in the death of anemones, two isogenic mutant strains of *V. parahaemolyticus* were used: CAB2 and CAB3. TDH/TRH genes for both mutant strains are deleted to exclude effects from these toxins and confounding with effectors. In addition, in CAB mutants, the transcription factors regulating the different structural proteins of the T3SS syringe were deleted. Specifically, in the CAB2 mutant, deletion of the *ExsA* transcription factor impairs regulation of T3SS1 genes (impT3SS1), while in the CAB3 mutant, the impaired T3SS2 gene expression was obtained by deletion of the *VtrA* transcription factor (impT3SS2) [22]. The contribution of T3SS in the mortality of *E. pallida* was assessed by exposing 12 anemones to different concentrations: 10^6^, 10^7^, and 10^8^ CFU/mL of *V. parahaemolyticus CAB2* and *CAB3.*

The dose–response experiment with these CAB mutants revealed distinct differences in morphological responses of the host to bacterial concentrations (Figure 2A). Anemones exposed to the CAB3 mutant at 10^6^ and 10^7^ CFU/mL developed clear signs of infection such as expulsion of gastrodermal cells containing algae, retraction of the column and tentacles, and extrusion of acontiae. In contrast, those exposed to the CAB2 mutant at 10^6^ and 10^7^ CFU/mL maintained extended tentacles, similar to the control animals not exposed to bacteria (Figure 2A).

These clear morphological differences were later correlated with dose-dependent mortality. Similarly to the WT strain (Figure 1B), the bacterial concentration of 10^8^ CFU/mL was lethal after 2 days independently of the mutant strain (Figure 2B,C). However, anemones exposed to CAB3 at 10^7^ CFU/mL showed 75% mortality by day 8 (Figure 2C,D), while the CAB2 mutant did not cause death over the same period (Figure 2B,D). Using the CAB3 mutant, which possesses an active T3SS1, we observed a similar lethality as with the WT strain, indicating that the injection of effectors by this syringe is largely responsible for the *E. pallida*’s mortality triggered by *V. parahaemolyticus*.

### 2.3. Expression of Bacterial Effectors in Presence of E. pallida

A key question was whether contact with *E. pallida* affects the expression of bacterial effector genes. Keeping in mind the role of secretion systems in injecting bacterial virulence effectors, here, we reveal the transcriptomic regulation of these effectors secreted by *V. parahaemolyticus* in contact with a whole sea anemone. Initial attempts consisted of quantifying effector gene expression by extracting RNA from bacteria infecting anemones, which yielded insufficient RNA. This limitation was overcome by incubating an individual anemone under constant agitation in a bacterial concentration sufficient for RNA extraction. Using RT-qPCR, we quantified the expression of four effector genes, three T3SS1 effectors: VPA0450, VopQ, and VopS, and one T3SS2 effector: VopC, after 1, 3, and 6 h in the presence of the anemone. In whole anemone treatment, the T3SS1 effector gene expression was significantly downregulated at 3 and 6 h compared with control bacteria without anemone (Figure 3A–C). In contrast, expression of the T3SS2 effector VopC was significantly upregulated after 6 h with the anemone (Figure 3D). These results demonstrated that the presence of the anemone has an impact on *V. parahaemolyticus* effector expression by specifically downregulating T3SS1 effectors and upregulating the T3SS2 VopC effector over a short period of a few hours.

### 2.4. Effectors of Gene Expression Induction via Molecular Cues

To investigate whether the observed changes in bacterial effector expression can be triggered by the sensing of molecules coming from anemones, we produced a purified anemone lysate to represent the chemical profile of the host. To only retain molecules, the anemone lysate preparation was filtered through a 0.22 μm filter. Importantly, the expression of T3SS1 and T3SS2 effectors was significantly upregulated after 3 and 6 h in the presence of lysate, relative to the controls (Figure 4). This expression was time-dependent since the exposure of WT *V. parahaemolyticus* to the lysate for 1 h was insufficient to induce changes in effector gene expression (Figure 4). The regulation of T3SS1 effectors expression in response to the lysate was completely reversed compared to that observed with whole anemone exposure (Figure 3A–C). Notably, both VPA0450 and VopQ were significantly over-expressed and maintained at elevated levels between 3 and 6 h (Figure 4A,B), while VopS showed an initial increase at 3 h, and a slightly lower level at 6 h (Figure 4C). In contrast, the level of expression of the T3SS2 effector VopC was significantly increased at 3 h and continued to increase after 6 h of exposure (Figure 4D).

These findings underscore the ability of *V. parahaemolyticus* to sense molecular cues from the anemone, leading to a differential regulation of its virulence factors.

## 3. Discussion

Our findings reveal that the zoonotic *V. parahaemolyticus* O3:K6 strain, widely recognized for its pathogenicity in human populations, also acts as an opportunistic pathogen in a marine cnidarian host. Specifically, its capacity to kill the sea anemone *E. pallida* at 27 °C offers strong evidence that human-derived strains of *V. parahaemolyticus* can retain virulence traits in environments beyond the human host and cause vibriosis in a sea anemone. In light of the increasing incidence of zoonotic *V. parahaemolyticus* in aquaculture, with notable impacts on shrimp farming causing diseases such as Acute Hepatopancreatic Necrosis Disease (AHPND) and vibriosis [23,24], our work highlights the pathogen’s broad ecological range and capacity for adaptation. Once introduced into coastal waters through anthropogenic activities, such as wastewater discharge, these clinically derived strains may exploit a range of virulence mechanisms to infect new hosts, thereby perpetuating a feedback loop between human populations and marine ecosystems.

Our results shed light on how the clinically derived, zoonotic *V. parahaemolyticus* O3:K6 strain exploits its range of virulence to infect a cnidarian host. We hypothesize that, once reintroduced into the ocean through human-associated pathways, these pathogenic strains capitalize on virulence traits—particularly those carried by different T3SS, to colonize and kill marine invertebrates including cnidarians. Our data confirm that T3SS1 plays a central cytotoxic role in this process, with effectors like VPA0450, VopQ, and VopS leading to host cell membrane disruption [25,26,27]. Critically, we show that T3SS1-driven cytotoxicity significantly contributes to the mortality of *E. pallida* over a week-long infection (Figure 2C), reinforcing the notion that certain T3SS1 effectors, long implicated in human infections, are equally potent in a cnidarian system.

Moreover, our findings highlight that the bacterium adjusts T3SS activity in a manner dependent on host contact duration. While T3SS1 effector gene expression is downregulated in the early hours of infection, T3SS2 VopC effector gene expression exhibits upregulation at 6 h (Figure 3). Studies have shown that VopC manipulates Rho GTPases to facilitate invasion of host cells [28,29]. Although not universally essential for pathogenicity, the elevated expression of VopC in the initial stages of infection in *E. pallida* suggests a potential role in breaching host cell barriers before the T3SS1 effectors exert their lethal cytotoxic effects.

In addition, our investigation into whether *V. parahaemolyticus* O3:K6 retains its ability to sense cnidarian hosts through chemosensing revealed a strong transcriptional increase in both T3SS1- and T3SS2-associated effectors when bacteria are exposed to host-derived cues (Figure 4). Similar chemosensing behavior has been observed in the coral pathogen *Vibrio coralliilyticus*, which detects coral-derived DMSP (dimethylsulfoniopropionate) and modulates its swimming behavior accordingly [30]. This capacity for chemical sensing emphasizes the ecological sophistication of *V. parahaemolyticus* in bridging the gap between human-associated environments and marine ecosystems.

From an ecological perspective, our findings amplify concerns about how zoonotic *V. parahaemolyticus* strains might pose an emerging threat to reef-building corals and other marine invertebrates. These strains, once adapted to humans through horizontal gene transfer, are able to persist in seawater and potentially disseminate among susceptible marine hosts. Under environmental pressures such as increasing sea surface temperatures and anthropogenic pollution, coral reefs are already highly vulnerable, and opportunistic infections by *Vibrio* species may further exacerbate reef decline.

Finally, the capacity of *V. parahaemolyticus* O3:K6 to infect both humans and cnidarians underlines its value as a model for comparative immunology. Like the human innate immune system, cnidarians rely on evolutionarily conserved pathways to detect and respond to microbial threats. By delineating how *E. pallida* recognizes and mounts defenses against T3SS effectors, we can better understand the ancestral origins of immunity and identify functional parallels to human host defenses. Our ongoing work aims to characterize these responses at the molecular and cellular levels and clarify the extent of shared immune pathways across metazoans. Taken together, these insights provide a foundation for further studies of pathogen evolution, host–pathogen interactions, and the interplay between marine and human health.

## 4. Conclusions

In conclusion, our study confirms that zoonotic *V. parahaemolyticus* can function as a cross-kingdom pathogen, demonstrating virulence in both human and cnidarian contexts. This accentuates the versatility of *V. parahaemolyticus* and warrants that vigilance is warranted to understand and mitigate the impacts of emerging pathogens in marine ecosystems. Future investigations should focus on elucidating the specific molecular interactions between *Vibrio* effectors and cnidarian immune sensors, as well as on environmental surveillance to assess the prevalence of zoonotic *V. parahaemolyticus* strains in coral reef habitats. By understanding these processes, we will be better equipped to predict disease outbreaks and protect both public health and delicate marine ecosystems.

## 5. Materials and Methods

### 5.1. Exaiptasia pallida Anemone Husbandry

Individual clones of the sea anemone *E. pallida* strain CC7 were acquired from the Pringle laboratory at Stanford University (Stanford, CA, USA) and kept at the Scientific Center of Monaco (Monaco, Monaco) under controlled conditions in the Ecosystems and Immunity Laboratory of the Biomedical Department. Anemones are maintained in three-liter tanks inside a culture chamber (Percival Scientific, Perry, IA, USA) at a constant 27 °C with 20 μmol·m^−2^·s^−1^ of light (12:12 h). Filtered seawater (FSW) at 0.22 μm is renewed every week and anemones are fed twice a week with artemia nauplii, except on the week of the experiment during which they fast. In preparation for the infection experiment, a single anemone clone or biological replicate of approximately half a centimeter in diameter was placed in each well of 24-well plates filled with 1 mL of 0.22 μm FSW. Anemones were kept still overnight until fixed to the bottom.

### 5.2. Bacterial Strain and Culture Preparation

*V. parahaemolyticus* strain RIMD 2210633 serotype O3:K6 came from the laboratory of Prof. Kodama (Osaka University, Japan). Mutant strains of *V. parahaemolyticus* (CAB2 and CAB3) serotype O3:K6 expressing GFP were provided by Prof. Kim Orth (UT Southwestern, Dallas, TX, USA).

Routinely, frozen *V. parahaemolyticus* are cultured at 37 °C in Luria–Bertani liquid medium (LB) supplemented with ampicillin for WT strain and spectinomycin for mutant strains and 3.5 g/L of NaCl for acclimatization to seawater salt concentration.

### 5.3. Infection Experiments and Mortality Assays

During the experiments, twelve anemones per concentration were challenged by balneation with three concentrations of the bacterium: 10^6^, 10^7^, or 10^8^ CFU/mL at 27 °C. The effect of the WT and mutant bacterial strains on the anemone’s fitness was recorded every day for eight days. The inoculation of the medium with bacteria was performed once on day 1. Anemone mortality assays consisted of recording the morphological response and degree of tissue damage and death by taking pictures under a binocular microscope (Stemi 305 trino, Zeiss, Oberkochen, Baden-Württemberg, Germany). Mortality was assessed by exposing the anemones to a quick exposition to bright light and touch stimuli, following the method described in Seneca et al. [20]. The absence of movement in response to both stimuli was recorded as death. The sub-lethal exposure was defined as the bacterial concentration sufficient to kill 50% of the anemones within a week period (LC50). From the infection experiments described above, mortality curves were generated by plotting the number of surviving anemones on each day for each bacterial concentration.

The results of the mortality assays were fundamental to determine the sub-lethal concentration used in subsequent experiments described below.

### 5.4. Production of E. pallida Lysate

Four anemones placed in a 1.5 mL tube filled with 1 mL of 0.22 μm FSW and 100 μL of 1 mm diameter glass beads were used for producing cell lysates. Tissue was completely disrupted using Precellys (Bertin Technologies, Montigny-le-Bretonneux, France) at 6500 bpm for 15 s. The lysate obtained was filtered through a 0.2 μm filter.

### 5.5. V. parahaemolyticus Virulence Stimulation

To conduct virulence stimulation assays, incubation of bacterial suspensions of 1 mL with 10^8^ CFU was put in contact with the whole anemone or anemone lysate for 1 h, 3 h, or 6 h at 27 °C with agitation. A volume of 100 μL of anemone cell lysate was added to different bacterial strain suspensions to subsequently compare their virulence states at the transcript level.

### 5.6. RNA Extraction

For bacterial extraction, bacterial cultures were pelleted by centrifugation at 6000× *g* for 5 min at 4 °C. The pellet was resuspended into 200 μL of Max Bacterial Enhancement Reagent (TRIzol Max Bacterial RNA Isolation Kit from Thermo Fisher Scientific, Waltham, MA, USA) and incubated at 95 °C for 4 min. A volume of 1 mL of TRIzol was then added to the reaction, and after an incubation of 5 min at room temperature, 200 μL of cold chloroform was added, and the reaction was hand-shaken for 15 s and let to settle at room temperature for 5 min. After centrifugation at 12,000× *g* for 15 min at 4 °C, the aqueous phase was then transferred to an equal volume of chilled 100% ethanol and moved onto a column for purification. The purification, including a DNase treatment step, followed the Directzol RNA MiniPrep Plus protocol (cat# R207 from Zymoresearch, Irvine, CA, USA). Total RNA was quantified at 260 nm wavelength using a Synergy H1M spectrophotometer. For anemone extraction, anemone tissues were lysed in Trizol reagent (Ambion life technologies, Waltham, MA, USA) to isolate RNA. One anemone was placed in a 1.5 mL tube filled with 1 mL of TRIzol and 100 μL of glass beads. Tissues were completely disrupted using Precellys (Bertin Technologies, Montigny-le-Bretonneux, France) at 6500 bpm for 15 s. Samples were then immediately placed on ice and total RNA extraction protocol was performed as in the “RNA extraction from bacteria” Section above.

### 5.7. Reverse Transcription and Quantitative Polymerase Chain Reaction (RT-qPCR)

The amount of RNA used in each reverse transcription reaction was 250 ng of anemone RNA or 1 μg of *V. parahaemolyticus* RNA in a total volume of 11 μL of nuclease-free water. Complementary DNA (cDNA) was produced using the RevertAid First Strand cDNA Synthesis (ThermosFisher Scientific, Waltham, MA, USA) according to the manufacturer’s instructions.

The RT-qPCR was performed using the Applied Biosystems (Foster City, CA, USA) Real-Time PCR instrument. The primers used are listed in Appendix A. The efficiency of all primers used in this study fell within a range between 95 and 100%. Each reaction of qPCR was conducted using 3 μL of cDNA added to 17 μL of the GoTaq^®^ qPCR Master Mix containing 10 μL of SYBR Green and 0.2 μL of CXR (Promega, Madison, WI, USA) for a 20 μL final volume in a 96-well plate. The PCR conditions were as follows: 2 min at 95 °C followed by 40 cycles of 15 s at 95 °C and 1 min at 60 °C, followed by a melt curve (60–95 °C). Six independent biological replicate samples were prepared per condition, samples were run in technical duplicates (mean values of technical duplicates were computed), and RT-qPCR data analysis was performed using the Pfaffl method [31]. The internal control gene used is RecA, a highly conserved gene involved in DNA repair and genetic recombination [32].

### 5.8. Statistical Analysis

Differences observed between infection experiments were statistically evaluated using Kaplan–Meier analysis performed to generate survival functions and log-rank significance values using GraphPad Prism 9.2.1. The results are expressed as the mean ± standard error. Statistical tests were performed by Analysis of variance (ANOVA) complemented by Welch’s test because we do not assume equal SDs. The significance was considered at a *p*-values < 0.05. All tests were performed on the software GraphPad Prism 9.2.1.

## Figures and Tables

**Figure 1 toxins-17-00175-f001:**
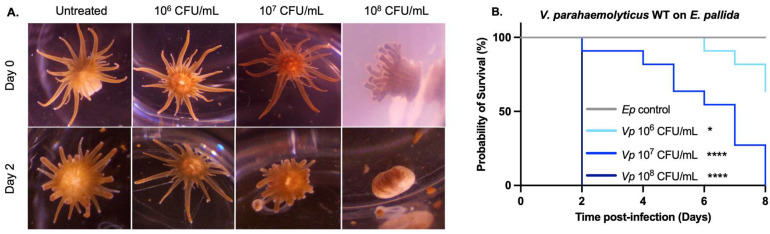
The LC50 for *E. pallida* exposed to *V. parahaemolyticus* is 10^7^ CFU/mL: (**A**) Morphological response of *E. pallida* exposed to live WT *V. parahaemolyticus* at three concentrations: 10^6^, 10^7^, and 10^8^ CFU/mL. The various stages of *E. pallida*’s response upon contact with WT *V. parahaemolyticus* are shown: shriveling of tentacles, increased mucus production, algae cells released, detachment, and liquification. (**B**) Survival curves of *E. pallida* exposed to live *V. parahaemolyticus* WT at different concentrations: 10^6^, 10^7^, and 10^8^ CFU/mL. Kaplan–Meyer test (****) meaning *p*-value < 0.0001 and (*) *p*-value < 0.05, *n* = 12.

**Figure 2 toxins-17-00175-f002:**
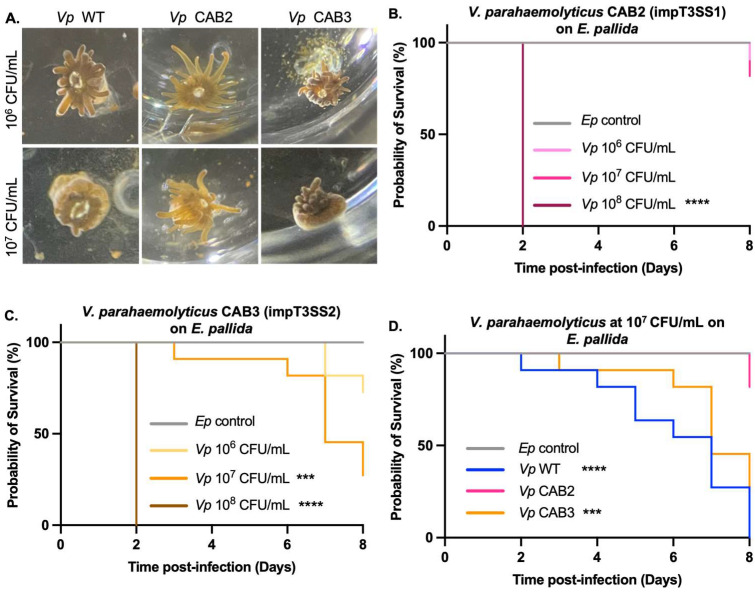
The CAB2 mutant relieved both morphological response and mortality observed under WT and CAB3 mutant: (**A**) Phenotype of *E. pallida* in balneation with *V. parahaemolyticus* (Vp): WT, CAB2, and CAB3. Photographs of *E. pallida* infected with WT, CAB2, and CAB3 strains at 10^6^ and 10^7^ CFU/mL, 7 days after infection. Anemones infected with the CAB2 strain have a phenotype similar to the uninfected control (elongated tentacles, light color) at both concentrations, while *E. pallida* infected with CAB3, and the WT strain, shows contracted tentacles and tissue lesions. (**B**,**C**) Mortality curves for *E. pallida* (Ep) in balneation with mutant strains of *V. parahaemolyticus* (Vp): CAB2 impaired T3SS1 (impT3SS1) (**B**), CAB3 impaired T3SS2 (impT3SS2) (**C**), for 8 days at 3 concentrations: 10^6^, 10^7^, and 10^8^ CFU/mL. (**B**) Significant decrease in survival of *E. pallida* exposed to CAB2 at 10^8^ CFU/mL. (**C**) Significant decrease in survival of *E. pallida* exposed to CAB3 at 10^7^ and 10^8^ CFU/mL. (**D**) Overlay of mortality curves of *E. pallida* (Ep) exposed to WT, CAB2, and CAB3 at 10^7^ CFU/mL. In the case of WT and CAB3, 50% of mortality is reached after 7 days. Kaplan–Meyer test (****) meaning *p*-value < 0.0001 and (***) *p*-value < 0.001, *n* = 12.

**Figure 3 toxins-17-00175-f003:**
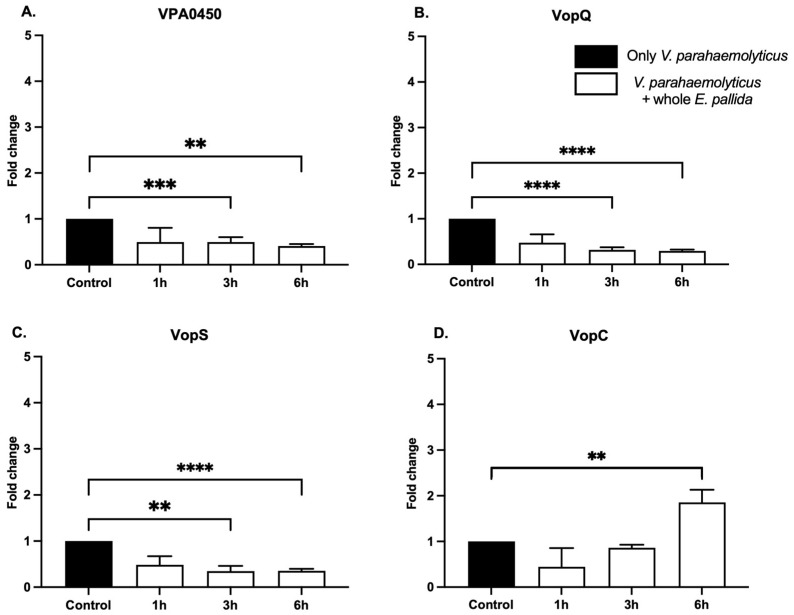
T3SS virulence factors of *V. parahaemolyticus* are transcriptionally regulated in the presence of whole *E. pallida*. Incubation of *V. parahaemolyticus* at 10^8^ CFU/mL in the presence of whole *E. pallida* (white) for 1 h, 3 h, and 6 h, and the control: only *V. parahaemolyticus* (black). Quantification of transcriptional regulation of *V. parahaemolyticus* virulence factors from T3SS1: VPA0450 (**A**), VopQ (**B**), VopS (**C**), and T3SS2: VopC (**D**). (**A**–**C**) Significant decrease in the number of *V. parahaemolyticus* T3SS1 virulence factors (VPA0450, VopQ, and VopS) transcripts after 3 h and 6 h incubation with whole *E. pallida*. (**D**) Significant increase in the number of VopC transcripts after 6 h incubation with whole *E. pallida*. T-test with Welch correction, meaning *p*-value < 0.0001 (****), < 0.001 (***) and *p*-value < 0.01 (**), *n* = 6.

**Figure 4 toxins-17-00175-f004:**
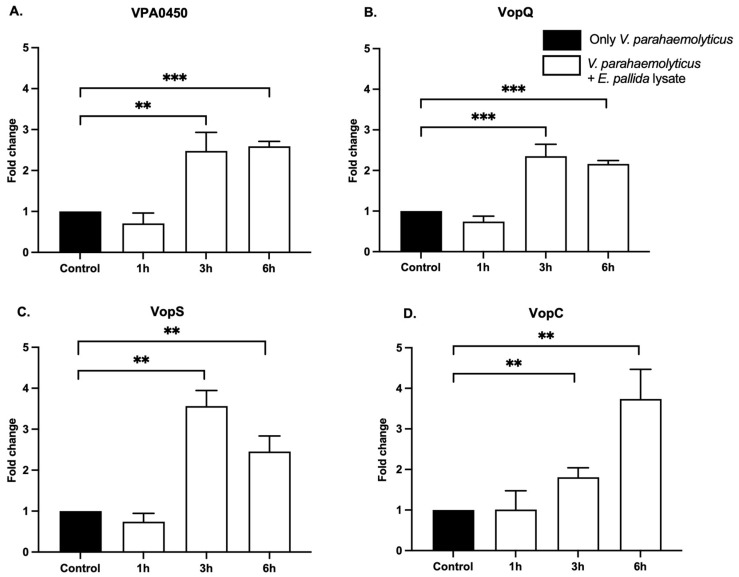
T3SS virulence factors of *V. parahaemolyticus* are transcriptionally upregulated in the presence of *E. pallida* lysate. Incubation of *V. parahaemolyticus* at 10^8^ CFU/mL in the presence of *E. pallida* lysate (white) for 1 h, 3 h, and 6 h, and the control: only *V. parahaemolyticus* (black). Quantification of transcriptional regulation of *V. parahaemolyticus* virulence factors from T3SS1: VPA0450 (**A**), VopQ (**B**), VopS (**C**), and T3SS2: VopC (**D**). (**A**–**D**). Significant increase in the number of transcripts of *V. parahaemolyticus* T3SS1 virulence factors (VPA0450, VopQ, and VopS), and T3SS2 virulence factor (VopC) after 3 h and 6 h incubation with *E. pallida* lysate. T-test with Welch correction, meaning *p*-value < 0.001 (***) and *p*-value < 0.01 (**), *n* = 6.

## Data Availability

The original contributions presented in this study are included in this article and Appendix A. Further inquiries can be directed to the corresponding authors.

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
