# Peer review of "Exaiptasia pallida Infection Model Reveals the Critical Role of Vibrio parahaemolyticus T3SS Virulence Factors in Its Pathogenicity for Sea Anemones"

_toxins, 2025, doi:10.3390/toxins17040175_

Round 1
Reviewer 1 Report
Comments and Suggestions for Authors
This manuscript, titled "Exaiptasia pallida Infection Model Reveals the Critical Role of Vibrio parahaemolyticus T3SS Virulence Factors in its Pathogenicity for Sea Anemones", presents an intriguing study on the pathogenicity of Vibrio parahaemolyticus towards the sea anemone Exaiptasia pallida. The research effectively fills an important knowledge void regarding the interaction between this bacterium and non - edible marine hosts. Moreover, it offers valuable perspectives on the role of the Type III Secretion System (T3SS) in this particular context.
Suggestions for Revision
1.Mechanistic Studies
Design and conduct experiments to delve into the molecular mechanisms underlying the interaction between T3SS effectors and anemone cells. Techniques such as confocal microscopy could be employed to visualize the localization of effectors within anemone tissues. Additionally, RNA - sequencing of anemone cells post - infection could be utilized to identify the downstream genes influenced by the action of these effectors.
2.Expanding the Experimental Scope
Incorporate different strains of E. pallida and V. parahaemolyticus in future experiments. Adopt an experimental setup similar to the current study, which includes determining the LC50, evaluating the function of T3SS, and analyzing the expression of effector genes. Furthermore, vary the environmental conditions during the infection experiments to simulate diverse natural habitats.
3.Enhanced Data Analysis
Apply supplementary statistical methods, such as Principal Component Analysis (PCA) and bootstrapping, as proposed above. In the revised manuscript, present the results of these analyses, accompanied by a discussion on how they supplement the existing statistical findings.
4.Consistency in Gene Name Representation
When describing genes, such as “T3SS1 effectors (VopS, VopQ, and VPA0450)”, there are occasional discrepancies in the order of gene names. While this does not hinder comprehension, maintaining a consistent order would render the presentation more meticulous. In scientific writing, where the consistency of terminology is highly emphasized, this aspect can be regarded as a minor shortcoming.
In summary, this is a promising study with great potential. By implementing the suggested improvements, the manuscript has the potential to make a more substantial contribution to the field of marine microbiology and host - pathogen interactions.
Author Response
We would like to thank Reviewer 1 for his/her insightful comments on our manuscript. We have considered all of them and addressed those we deemed relevant to this particular manuscript. The reason we could not address some of the comments in this manuscript is that the results are being reported in another manuscript focusing on the innate immune response of the anemone. Please find below our response to each comment:
1. Experiments to visualize localization of GFP-labelled Vibrio in anemone host tissues using confocal microscopy were previously conducted in our laboratory and reported elsewhere (Billaud et al. 2023). We have cited this work in this manuscript.
We recently conducted a RNASeq experiment on the transcriptomic response of anemones infected with wild-type or VopC effector deleted mutant Vibrios. The comparison of those responses identifies the downstream genes influenced by the action of VopC during infection as suggested by Reviewer 1. The results of this study are being reported in a different manuscript in preparation.
2. We agree with Reviewer 1 that the use of different strain of Vp, especially a non pathogenic environmental strain could be a very valuable way to evaluate the role of T3SS in the future. Varying the environmental conditions is also something that we intend to do in the future. Thank you for this suggestion.
3. PCA and bootstrapping are some of the statistical analyses we have applied to our RNASeq data on the anemone’s response to Vibrio infection. We hope that the Reviewer 1 will find those results interesting when they will be published.
4. We have reviewed and double-checked nomenclature discrepancies throughout the manuscript. We agree with Reviewer 1 on this important aspect of scientific writing and believe that it is a definitive improvement to our manuscript.
Reviewer 2 Report
Comments and Suggestions for Authors
Overall, the research work is carried out correctly, and the approaches to finding the answers to the questions are correct. Some corrections and suggestions will be made to improve the work, especially in the methods section.
Table S1. Set the directionality of the primer oligonucleotides.
Figure S1. The full name of the microorganisms should be provided at least once.
Lines 133-136: This experiment is very important since it deals with the complete anemone. When the lysate is used, it is expected that bacterial effectors will be overexpressed. Therefore, I think the interaction with the complete anemone is more relevant. So, this method could be better described, and a way could be found to remove the bacteria from the interaction with the host or to have it in greater quantity to verify the expression of its effectors.
Lines 182-183: Homeostatic dysregulation of ions or the cytoskeleton was not demonstrated in this study, so I suggest leaving these lines until cell membrane disruption.
249: What is the selective antibody for each strain?
254: check the correct writing of the units.
259-262: Reference of this method.
268: check the correct writing of the units.
303-305: better description of this method.
What is the internal control gene used? In the experiments on the expression of bacterial effectors.
Author Response
We would like to thank Reviewer 2 for his/her helpful editing and comments on our manuscript. We have considered all of them and addressed them in red font in the manuscript.
Table S1. The directionality of the primers is now shown.
Figure S1. The full name of the microorganisms is provided.
Lines 133-136. We have made changes to better describe our method. We take this comment into consideration and will keep it in mind in future experiement.
Lines 182-183. We understand and agree with Reviewer 2. We removed the problematic lines.
Line 249. We added the information about the selective antibody.
Line 254. We corrected the writing of the unit.
Line 259-262. We added the reference.
Line 268. We corrected the writing of the unit.
Line 303-305. We have provided a better description of this method, and the information of the internal control gene for qPCR on bacterial effectors.